# Carbon–nitrogen transmutation in polycyclic arenol skeletons to access *N*-heteroarenes

Hong Lu [1,3], Yu Zhang[1,3], Xiu-Hong Wang[1], Ran Zhang[1], Peng-Fei Xu [2] ✉ & Hao Wei [1] ✉

Developing skeletal editing tools is not a trivial task, and realizing the corresponding single-atom transmutation in a ring system without altering the ring size is even more challenging. Here, we introduce a skeletal editing strategy that enables polycyclic arenols, a highly prevalent motif in bioactive molecules, to be readily converted into *N*-heteroarenes through carbon–nitrogen transmutation. The reaction features selective nitrogen insertion into the C–C bond of the arenol frameworks by azidative dearomatization and aryl migration, followed by ring-opening, and ring-closing (ANRORC) to achieve carbon-to-nitrogen transmutation in the aromatic framework of the arenol. Using widely available arenols as *N*-heteroarene precursors, this alternative approach allows the streamlined assembly of complex polycyclic heteroaromatics with broad functional group tolerance. Finally, pertinent transformations of the products, including synthesis complex biheteroarene skeletons, were conducted and exhibited significant potential in materials chemistry.

Organic synthesis underpins the evolution and advancement of broad areas of science, from materials to medicine. Arenes are among the most widely used rings in medicine and natural products. The functionalization of arenes is a particularly attractive tool for the production of pharmaceuticals, natural products, and molecular materials[1–4]. However, their application has so far been largely focused on C–H functionalization chemistry (peripheral editing), and the precise modification of the aromatic ring skeleton remains largely unexplored (Fig. 1A)[5–7]. Single-atom skeletal editing has become an extremely powerful tool for straightforwardly modifying the core skeleton of organic molecules. Recently, a limited number of single–atom insertion or deletion reactions have been developed to reshape the underlying molecular skeletons[8–21]. However, the direct modification of valuable core structures by replacing one atom in a ring system without changing the ring size and aromaticity remains elusive[22–31], although it has been recognized as a highly desirable transformation.

One of the most studied among *N*-heteroarenes are pyridines, which serve as a bioisosteric replacement of benzene counterparts

within the parent molecules[32–34]. The replacement of carbon with nitrogen in aromatic ring systems can have several important effects on the molecular and physicochemical properties relevant to multiparameter optimization (Fig. 1B). This necessary nitrogen atom effect is a versatile high-impact design element for multiparameter optimization, which has been shown to improve various key pharmacological parameters[35]. Recently, Burns and Levin independently reported groundbreaking methods for the direct conversion of arenes to pyridines via nitrene internalization (Fig. 1C)[36,37]. In these process, additional steps for installation and isolation of aryl azides are always requried, which indicated that a selective, and straightforward transformation of diverse arenes into *N*-heteroarenes remains an important goal[38].

A key challenge in this transformation is the stability of the aromatic compounds. Our design overcomes this intrinsic challenge using arenols as substrates. Dearomatization of arenols disrupts the stability of the aromatic ring and promotes subsequent skeletal transformations[39–48]. Arenol can also act as a selectivity controlling element in site-selective skeletal editing. Our group's recent work

[1]Key Laboratory of Synthetic and Natural Functional Molecule of the Ministry of Education, College of Chemistry & Materials Science, Northwest University, Xi'an 710069, China. [2]State Key Laboratory of Applied Organic Chemistry, College of Chemistry and Chemical Engineering, Lanzhou University, Lanzhou 730000, China. [3]These authors contributed equally: Hong Lu, Yu Zhang. ✉e-mail: xupf@lzu.edu.cn; haow@nwu.edu.cn

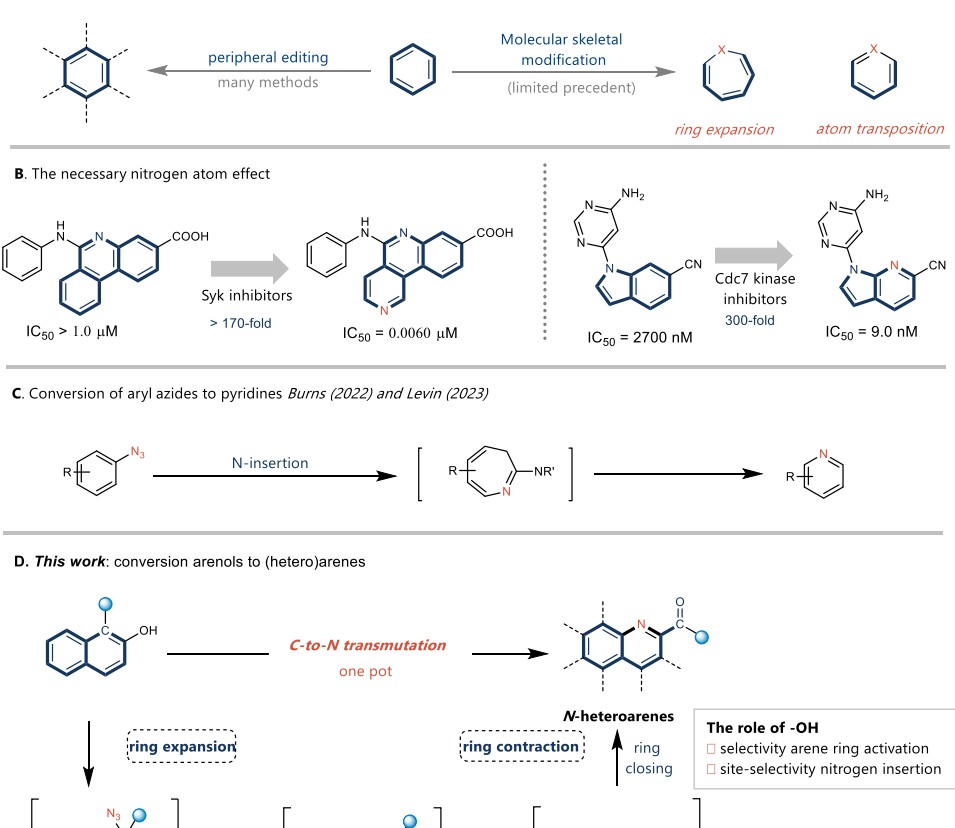

**Fig. 1 | Examples of carbon–nitrogen transmutation and our reaction design. A** Molecular editing of aromatic rings. **B** Examples of the necessary nitrogen atom effect. **C** Conversion of aryl azides to pyridines. **D** This study.

## Table 1 | Screening of reaction conditions.[a]

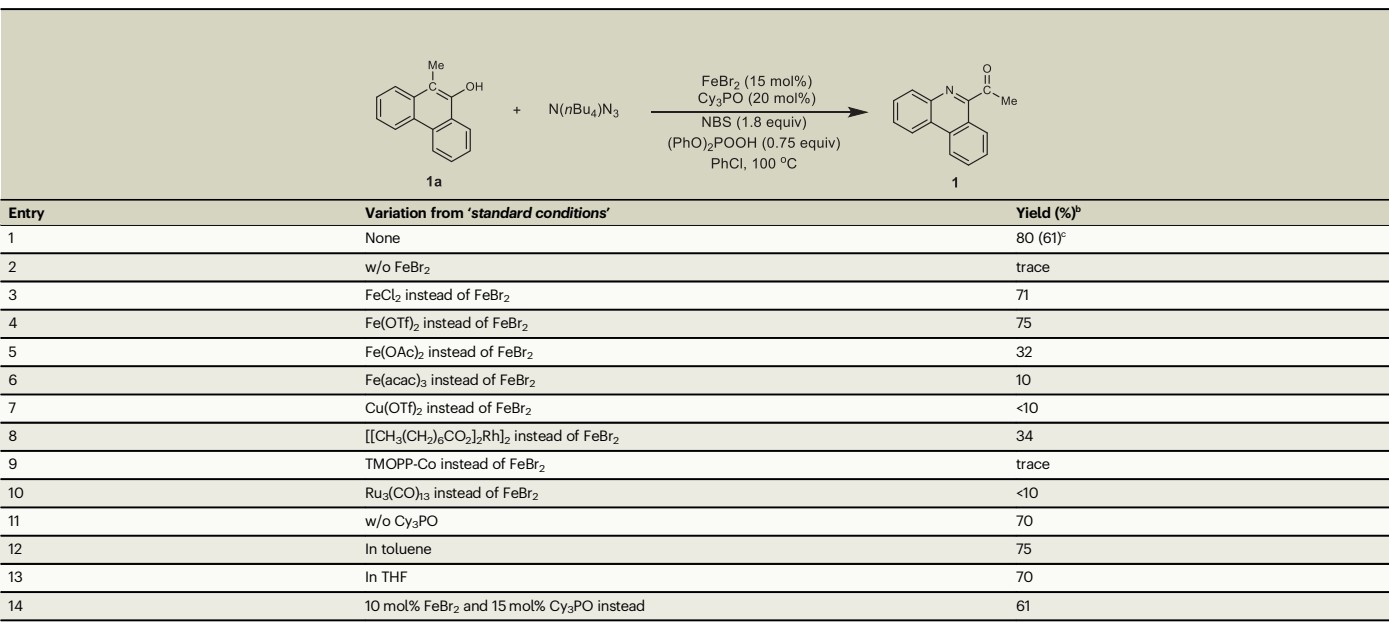

| Entry | Variation from 'standard conditions' | Yield (%)[b] |
|---|---|---|
| 1 | None | 80 (61)[c] |
| 2 | w/o FeBr$_2$ | trace |
| 3 | FeCl$_2$ instead of FeBr$_2$ | 71 |
| 4 | Fe(OTf)$_2$ instead of FeBr$_2$ | 75 |
| 5 | Fe(OAc)$_2$ instead of FeBr$_2$ | 32 |
| 6 | Fe(acac)$_3$ instead of FeBr$_2$ | 10 |
| 7 | Cu(OTf)$_2$ instead of FeBr$_2$ | <10 |
| 8 | [[CH$_3$(CH$_2$)$_6$CO$_2$]$_2$Rh]$_2$ instead of FeBr$_2$ | 34 |
| 9 | TMOPP-Co instead of FeBr$_2$ | trace |
| 10 | Ru$_3$(CO)$_{13}$ instead of FeBr$_2$ | <10 |
| 11 | w/o Cy$_3$PO | 70 |
| 12 | In toluene | 75 |
| 13 | In THF | 70 |
| 14 | 10 mol% FeBr$_2$ and 15 mol% Cy$_3$PO instead | 61 |

[a]Unless otherwise specified, all reactions were carried out using **1a** (0.1 mmol), NBS (0.18 mmol), N($n$Bu$_4$)N$_3$ (0.3 mmol), (PhO)$_2$POOH (0.075 mmol), FeBr$_2$ (0.015 mmol), and Cy$_3$PO (0.02 mmol) in PhCl (1.0 mL) at 100 °C for 36 h.
[b]Isolated yields after chromatography.
[c]Scale-up reaction by using 1.0 mmol of **1a**.

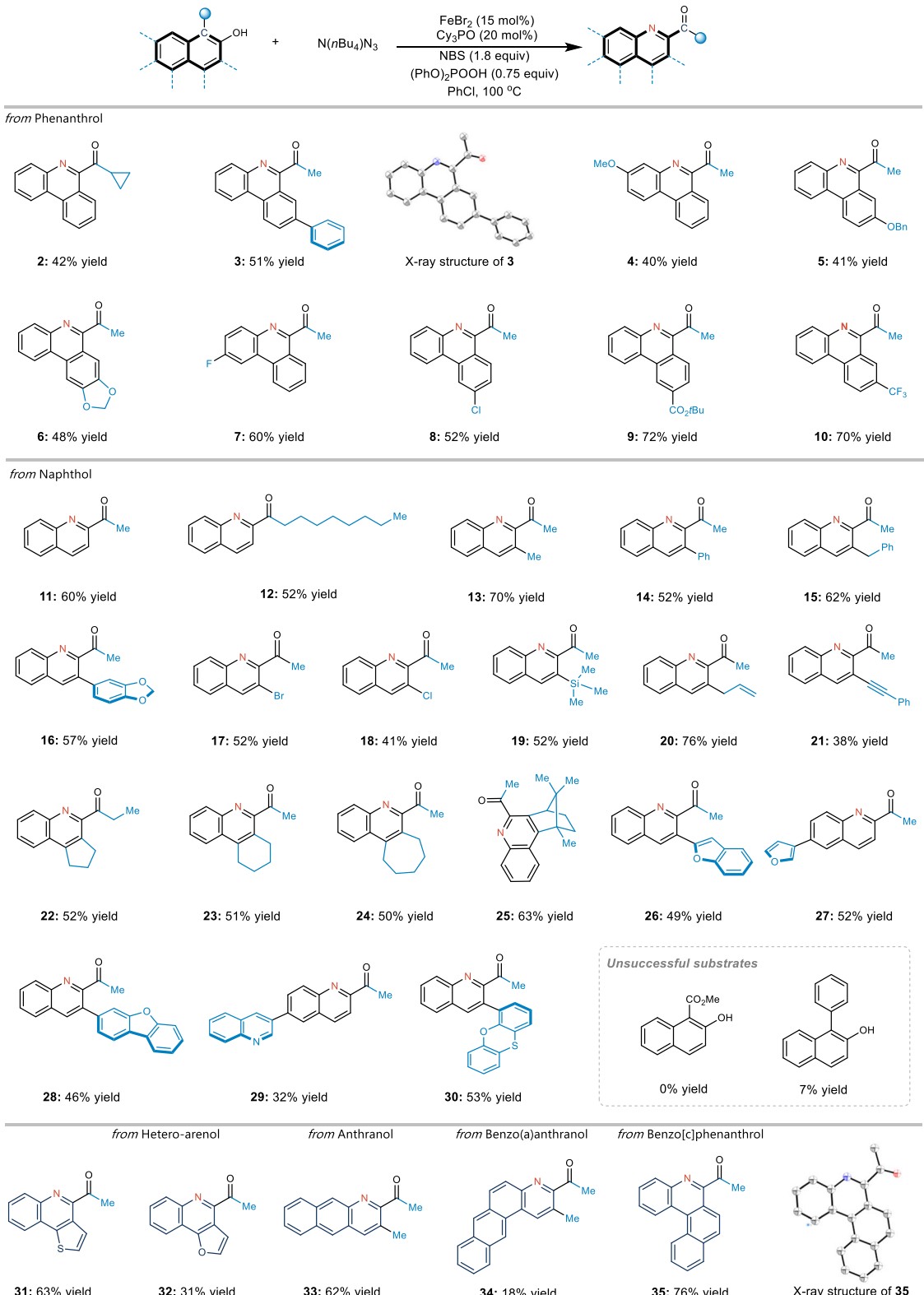

**Fig. 2 | Substrate scope of the carbon–nitrogen transmutation in polycyclic arenol.** [a,b] [a]Isolated yields after chromatography are shown. [b] Reaction conditions: substrate (0.1 mmol), NBS (0.18 mmol), N($n$Bu)$_4$N$_3$ (0.3 mmol), (PhO)$_2$POOH (0.075 mmol), FeBr$_2$ (0.015 mmol), and Cy$_3$PO (0.02 mmol) in PhCl (1.0 mL) at 100 °C for 36 h.

employing this dearomative strategy to promote ring expansion of arenols inspired us to continue investigating this strategy to more complex skeletal editing transformation[49]. In this work, we describe the direct carbon-to-nitrogen transmutations in arenols. This reaction involves two stages: ring expansion and contraction (Fig. 1D). In the first stage, the insertion of nitrogen atoms is achieved by azidative dearomatization of an arenol and intramolecular aryl migration. In the second stage, a carbon atom moves out of the ring skeleton through ring-opening, and ring-closing (ANRORC), which ultimately furnishes

**Fig. 3 | Application potential of carbon–nitrogen transmutation. A** Applications that allow access to complex biheteroarene skeletons. **B** Preparation of unconventionally 3,6-substituted quinolines. **C** Sequential skeletal editing transformations of naphthol.

desired carbon–nitrogen transmutation in polycyclic arenol skeletons[50–52].

## Results

### Reaction optimization

We began our investigation using methylphenanthren-9-ol **1a** as the reaction partner. (PhO)$_2$POOH, NBS, and N($n$Bu)$_4$N$_3$ were employed as reagents for the in situ formation of the azido ketone intermediate (see Supplementary Information, section 2.2.2). For optimization, we observed the formation of desired product **1** in 80% yield using FeBr$_2$ and Cy$_3$PO as an effective catalyst–ligand combination in PhCl (Table 1, entry 1). A control experiment revealed that an iron salt was essential for obtaining the desired product (Table 1, entry 2). Other iron salts, including FeCl$_2$, Fe(OTf)$_2$, Fe(OAc)$_2$, and Fe(acac)$_3$, exhibited lower efficiency than inexpensive FeBr$_2$ (Table 1, entries 3 – 6). Furthermore, when other established metal nitrenoid formation catalysts, including copper, rhodium, cobalt, and ruthenium, were used, the desired product was not obtained satisfyingly (Table 1, entries 7–10)[53,54]. Further optimization showed that this reaction was slightly improved using Cy$_3$PO (Table 1, entry 11). The reaction appeared to be less sensitive to solvents, as replacing the PhCl with either toluene or THF furnished **1** in good yield (Table 1, entries 12 and 13). The yield

decreased slightly when 10 mol% FeBr$_2$ and 15 mol% Cy$_3$PO were used (Table 1, entry 14).

### Substrate scope

Considering the optimal reaction conditions, the substrate scope was determined (Fig. 2). Various arenols, including phenanthrol (**1–10**), naphthol (**11–30**), anthranol (**33**) benzo($a$)anthranol (**34**), and benzo[$c$]phenanthrol (**35**), can effectively undergo the desired carbon–nitrogen transmutation. Both electron-rich and electron-deficient aromatic substrates were suitable for the process. It was found that the substituents at the *ortho* positions of the arenol are significant. When the substituent was an alkyl group, the corresponding arenols underwent atom transmutation smoothly in moderate-to-good yield and chemoselectivity. The presence of a phenyl group or an electron-withdrawing group such as CO$_2$Me at the *ortho*-position can inhibit this reaction. However, various functional groups, such as ether (**4** and **5**), acetals (**6** and **16**), aryl halides (**7, 8, 17** and **18**), esters (**9**), trifluoromethyl (**10**), trimethylsilyl (TMS) (**19**), alkenes (**20**), and alkynes (**21**), were tolerated in this transformation. In addition, several naphthyl-fused rings (**22–25**) were suitable substrates, affording the desired products in moderate-to-good yields. Heterocyclic moieties such as benzofuran (**26**), furan (**27**),

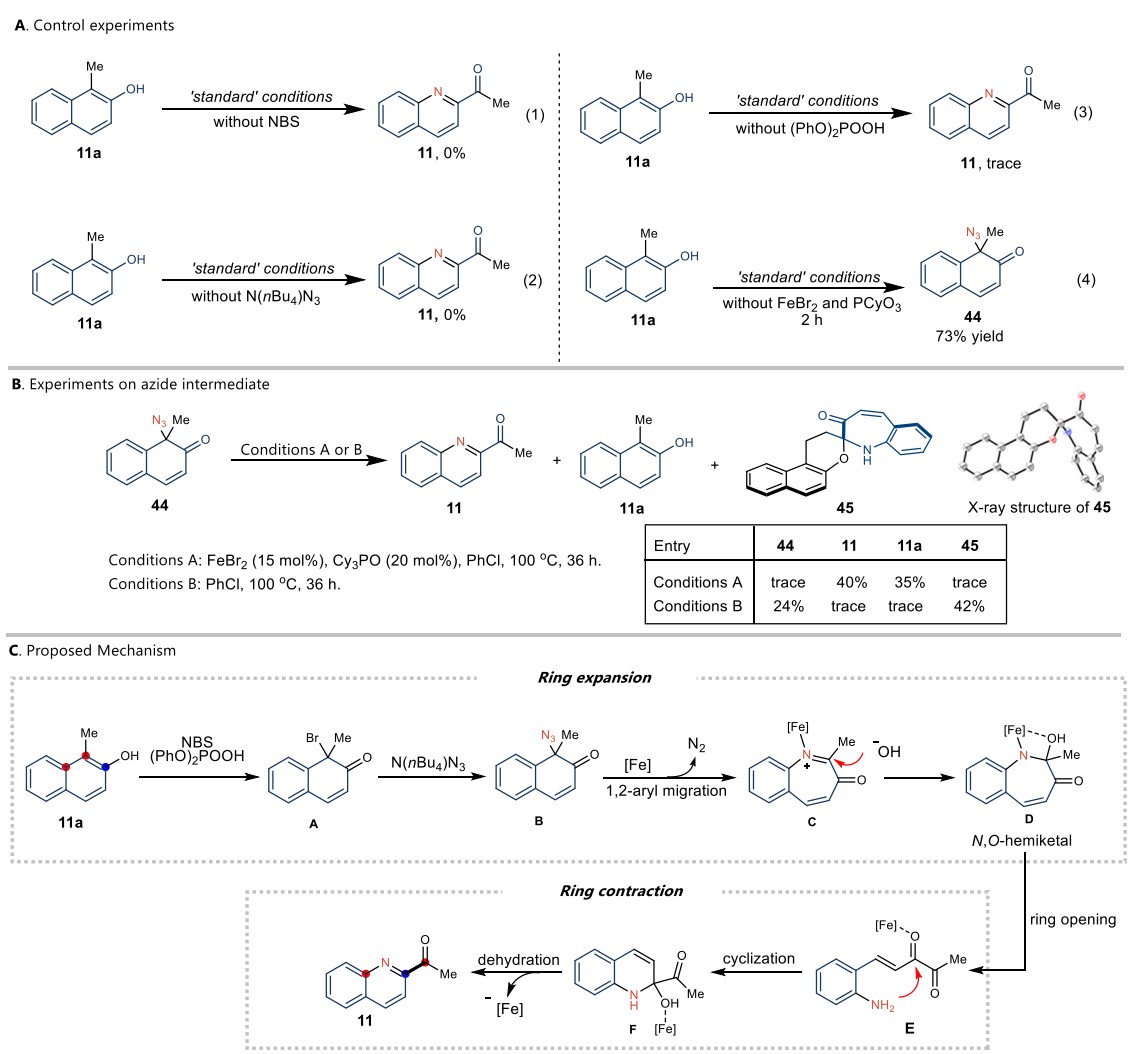

**Fig. 4 | Mechanistic studies. A** Control experiments. **B** Control experiments using azide intermediate showing that Fe catalyst is essential for aryl migration and ring contraction process. **C** Proposed mechanism.

dibenzofuran (**28**), quinoline (**29**), and phenoxathiine (**30**) were also compatible. Moreover, fused heteroarenes such as naphtho[1,2-*b*]thiophene (**31**) and naphtho[1,2-*b*]furan (**32**) can be incorporated, providing pharmaceutically interesting fused-ring skeletons that are non-trivial to prepare. The structures of **3** and **35** were identified using X-ray crystallography.

## Synthetic utility

The successful development of the atom transmutation protocol offers a rapid and modular approach to access complex biheteroarene skeleton, a common structural motif found in bioactive compounds (Fig. 3A). Compound **37** could be easily transformed to the iridium complexes **39**, which could serve as the red-light-emitting electrochemical cell[55,56]. Next, the synthetic versatility of the C-to-N transmutation was demonstrated through the preparation of 3,6-disubstituted quinolines **42**, which could not be obtained from directly electrophilic substitution of quinolines (Fig. 3B). Specifically, the successful development of the carbon–nitrogen transmutation offers exciting opportunities to devise more complex skeletal editing transformations via combinations of atom insertions and deletions. Benzo[1,4]diazepine **43** can be accessed through a C–H azidation and aryl migration sequence from **11**, presently the formal carbon deletion and two nitrogen insertion products of starting naphthol **11a** (Fig. 3C).

## Mechanistic considerations

To elucidate the mechanism of this transformation, series control experiments were first conducted. The reactions without addition of NBS or N($n$Bu$_4$)N$_3$ failed to produce the desired product (Fig. 4A, equations 1 and 2). And trace amount of **11** was observed when (PhO)$_2$POOH was absent from the reaction mixture (Fig. 4A, equation 3). It's worth noting that azide ketone **44** could be isolated in 73% yield in the absense of Fe catalyst after 2 h (Fig. 4A, equation 4). These results indicated that the proposed azidative dearomatization of arenol might be involved (Fig. 1D)[57]. The azide ketone **44** was then tested with and without the addition of the Fe catalyst (Fig. 4B). It was found that the desired product **11** was formed in 40% yield, and 35% yield of **11a** was isolated in presence of Fe catalyst, which demonstrated that the proposed azidation is a reversible process via successive single-electron transfer (SET) from Fe(II) to eliminate azide[58]. On the contrary, only trace amount of product **11** and **11a** was observed without Fe catalyst. And a byproduct **45** was detected in 42% yield and recovered **44** in 24%[59]. These results revealed that the Fe catalyst is not only involved in aryl migration, but is also essential for the ring contraction process[60].

Based on the literature reports and our observations, a plausible mechanism is proposed (Fig. 4C). Initially, the *N*-bromosuccinimide-mediated dearomatization of the corresponding naphthol of **11a** afforded the brominated ketone intermediate **A**, which subsequently

reacted with N($n$Bu)$_4$N$_3$ to generate azido ketone **B**. Then iron salt reacted with **B** can form metal−nitrene species, which would then undergo 1,2-aryl migration to form ring expansion intermediate **C**. Subsequently, addition of hydroxide anion to the imine group of **C** induces $N,O$-hemiketal **D**. The collapse of **D** with assistance of iron salt produces ring-opening amino-ketone intermediate **E**[61,62], which undergoes re-cyclization and dehydration to form stable benzoquinoline **11** and release Fe catalyst.

In conclusion, this study proposed a unique strategy that enables straightforward carbon-to-nitrogen transmutations in arenols through a one-pot ring expansion-contraction sequence. This site-selective atom transformation is based on sequentially combining three transformations in one pot using aryl migration and imine transposition as key steps and opens new opportunities for single-atom skeletal edit design. Further preparation of complex biheteroarene skeleton and unconventionally substituted quinoline highlights the potential of this study. This provides an alternative for the development of $N$-heteroarenes and demonstrates significant potential in materials chemistry.

## Methods

### General condition for carbon−nitrogen transmutation

Substrate (0.1 mmol), NBS (0.12 mmol), N($n$Bu$_4$)N$_3$ (0.2 mmol), FeBr$_2$ (0.015 mmol), tricyclohexylphosphine oxide (0.02 mmol), (PhO)$_2$POOH (0.05 mmol), and PhCl (1.0 mL) was successively added to an 10 mL sealed tube equipped with a Teflon-coated magnetic stir bar. The tube then was sealed with a Teflon screw cap and placed on a hotplate pre-heated to 100 °C with vigorous stirring. After 18 h, the reaction was cooled to room temperature and another portion of NBS (0.06 mmol, 0.6 equiv), N($n$Bu$_4$)N$_3$ (0.1 mmol) and (PhO)$_2$POOH (0.025 mmol) was successively added to the sealed tube. The tube then reacted at 100 °C with vigorous stirring. After 18 h, the reaction was cooled to room temperature. The solvent was evaporated and the residue was directly purified by flash column chromatography on silica gel (petroleum ether/ethyl acetate = 20/1) to give the desired products.

## Data availability

Data relating to the optimization studies, mechanistic studies, general methods, and the characterization data of materials and products, are available in the Supplementary Information. Crystallographic data for the structures reported in this article have been deposited at the Cambridge Crystallographic Data Center, under deposition numbers CCDC 2285580 (**3**), 2285872 (**35**), 2310917 (**39**) and 2308629 (**45**). Copies of the data can be obtained free of charge via https://www.ccdc.cam.ac.uk/structures. All data are available from the corresponding author upon request.

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

## Acknowledgements

We are grateful for the financial support from the National Natural Science Foundation of China (22271231, H.W.), Key Research and Invention Program in Shaanxi Province of China (2021SF-299, H.W.), Natural Science Basic Research Program of Shaanxi (2020JQ-574, H.L.), Scientific Research Program of Shaanxi Education Department (20JK0937, H.L.) and Northwest University.

## Author contributions

H.W. conceived and designed the project and composed the manuscript. H.L., Y.Z., X.W., and R.Z. conducted the experiments and analyzed the data. H.L. and P.X. discussed the experimental results and commented on the manuscript. H.W. conducted general guidance, project directing, and manuscript revisions.

## Competing interests
The authors declare no competing interests.

## Additional information

**Peer review information** : *Nature Communications* thanks Yuhua Deng and the other anonymous reviewer(s) for their contribution to the peer review of this work. A peer review file is available.

