## [Peer Review File · Nature Communications]

Carbon–Nitrogen Transmutation in Polycyclic Arenol Skeletons to Access N-HeteroarenesREVIEWER COMMENTS

Reviewer #1 (Remarks to the Author):

A method for carbon–nitrogen transmutation in polycyclic arene is reported in this manuscript. This reviewer believes that it would be more suitable for a more specialized journal such as *The Journal of Organic Chemistry* for the following reasons.

[1] The present work does not possess significant novelty.

The chemistry of the current transformation bears considerable similarity to that of the reaction reported by the authors (ref. 35). The conversion of arenes into intermediate II through I, as described in Scheme 1, has been previously documented. However, the authors do not mention this in the text at all, despite providing a citation number in passing. This represents an unfair writing style, potentially construed as an attempt to claim excessive novelty by obscuring the extent of existing knowledge.

Furthermore, a similar conversion from II to III was also reported.

-*Heterocycles* 1981, 363.

-*Tetrahedron Lett.* 2015, 6886.

Although the reaction conditions and substrates may differ somewhat, the conversion of azepine to pyridine is not particularly surprising.

Overall, the progress of this study lies in identifying a system wherein two previously known reactions occur consecutively. This reviewer does not consider this result to be sufficiently novel or significant to merit publication in *Nat. Commun.*

[2] Citations are not appropriate.

The authors appear overly focused on citing high-impact factor articles, neglecting relevant ones. Neither the *Heterocycles* nor *TL* papers mentioned above are cited.

Additionally, the authors stated, “However, the direct modification of valuable core structures by replacing one atom in a ring system without changing the ring size and aromaticity remains elusive, although it has been recognized as a highly desirable transformation”.

The direct modification of valuable core structures by replacing one or two atoms in a ring system without changing the ring size and aromaticity remains has been reported continuously in recent years. The present study is deeply interconnected with this literature and should be cited.

- *J. Am. Chem. Soc.* 129, 12640

- *Adv. Synth. Catal.* 355, 2495–

- *Chem. Commun.* 55, 8575

- *Org. Lett.* 23, 6126

- *Chem. Sci.* 14, 485

- *Nat. Chem.* doi.org/10.1038/s41557-023-01428-2

[3] The quality of some data is unacceptable.

The following NMR spectra exhibit significant levels of by-product contamination.

Consequently, they may overestimate yields. It is recommended to purify the product sufficiently to obtain a clean spectrum before determining the yield, or alternatively, quantify the impurities via GC or other means to obtain accurate yields.

Compounds 15, 16, 19(DCM?), 20, 38 (EtOAc?)

Reviewer #2 (Remarks to the Author):

In the proposed manuscript, authors describe an elegant skeletal editing strategy involving a straightforward carbon-to-nitrogen transmutations in arenols through a one-pot ring expansion-contraction sequence. This strategy avoided the dangers and laborious preparation of aryl azides as raw material, while using the readily available arenols as substrate with N(nBu)₄N₃ as reagent is a skillful strategy. I believe this report will inspire more researchers to develop other more perfect skeletal editing protocols and it will be bound to draw more researchers to apply the skeletal editing. In my own opinion, the scientific novelty of the manuscript is very high, and the quality of the work is high. My only critique is about the scale-up synthesis. I am satisfied with it too. My recommendation is to publish the manuscript.

Reviewer #3 (Remarks to the Author):

The manuscript authored by Xu, Wei, and co-workers presents a novel approach for carbon-nitrogen transmutation in polycyclic phenols to synthesize N-heteroarenes. The method entails the conversion of 10-alkylphenanthren-9-ols and 1-alkylnaphthalen-2-ol-based starting materials into 6-acyl phenanthridine and 2-acyl quinolines, respectively. Employing a Fe-catalyst with a phosphine oxide ligand alongside NBS, TBAN₃, and diphenylphosphonate in reagent quantities distinguishes this reaction from a previously reported protocol utilizing a Cu-catalyst, which produced benzazepine products from methyl 2-hydroxy-1-naphthoate as a starting material. The strategic selection of starting materials induces a cascade reaction culminating in the formation of aromatic products. The manuscript details the exploration of scope of this reaction, entailing various substitutions around the ring (except the ortho position of the arenol). The authors describe a number of strategies for conveying this reaction into useful synthetic sequences (both in product elaboration and redesign of synthetic sequences to use arenol chemistry for diversification). Mechanistic elucidation, derived from control experiments and intermediate observations, underscores the sequencing of steps within the cascade reaction.

This reaction is sufficiently novel and useful to warrant publication of this manuscript in Nature Communications. However, a small number of revisions (with a couple of additional experiments to conduct) should be carried out prior to publication.

Specific comments:

1. The manuscript shows quite a bit of scope, except in the ortho substituent. The extent of scope in this regard is product 2 and product 12, which swap methyl for cyclopropyl and n-octyl. The authors should add a small set of additional substrates to explore the scope outside simple alkyl groups (Ph, CO₂Me, etc). Even if the results are negative, this would be informative to readers in terms of utility of this reaction.
2. The authors should comment more clearly on the reaction of 44 to give a roughly 1:1 mixture of product 11 and deazidated 11a – what is the pathway of the deazidation?
3. There are repeated typos referring to tricyclohexylphosphine oxide as tricyclohexylphosphine, both in the methods section of the manuscript and in the SI.
4. Additional referencing to N-atom insertion of nitro groups into arenes, accessing structurally similar 7-membered intermediates as in this paper, should be included. Work

from Leonori (10.1002/anie.202310540, 10.1038/s41557-023-01429-1) and Radosevich (10.1021/jacs.2c12450) come to mind. The early work of Sundberg (10.1021/ja00757a032, 10.1021/jo00828a067, 10.1021/ja01031a024) should also be cited for appropriate context.

Responses to the reviewer 1's comments

- 1 *The present work does not possess significant novelty. The chemistry of the current transformation bears considerable similarity to that of the reaction reported by the authors (ref. 35). The conversion of arenols into intermediate II through I, as described in Scheme 1, has been previously documented. However, the authors do not mention this in the text at all, despite providing a citation number in passing. This represents an unfair writing style, potentially construed as an attempt to claim excessive novelty by obscuring the extent of existing knowledge. Furthermore, a similar conversion from II to III was also reported. -Heterocycles 1981, 363. -Tetrahedron Lett. 2015, 6886. Although the reaction conditions and substrates may differ somewhat, the conversion of azepine to pyridine is not particularly surprising. Overall, the progress of this study lies in identifying a system wherein two previously known reactions occur consecutively. This reviewer does not consider this result to be sufficiently novel or significant to merit publication in Nat. Commun.*

Response: We really appreciate the reviewer for the comments on our manuscript. We fully agree with the reviewer's opinion. I believe it may be due to the insufficient description in the introduction section that the innovative aspects of our article were not fully reflected. Therefore, we have revised the introduction section accordingly. Moreover, we have also detailed the relationship between this work and our previous work (ref. 35, now ref. 49). Thank you for the reviewer's reminder.

Additionally, this manuscript is of broad interest to the readership of *Nature Communications* for the following reasons:

1. Although our previous work was also able to achieve carbon-to-nitrogen transmutation (ref. 35, now ref. 49), the reaction required multi steps and involved the use of strongly oxidizing conditions. The development of single-step (or single-operation) transformations is imperative for the broad adoption of single-atom editing. The previous work suffered substantial yield losses when telescoped to a single operation due to competitive reactivity with by-products. However, this work overcomes these shortcomings. The reaction is not only completed in a single step but also avoids the use of strong oxidizing agents. Importantly, the mechanism of the ring contraction process is completely different from that of our previous work.
2. Although a similar conversion from II to III was also reported (-*Heterocycles* 1981, 363.-*Tetrahedron Lett.* 2015, 6886), their mechanism and the applicable substrate scope are not the same as this reaction. They are not suitable for this complex carbon-to-nitrogen transmutation process.

- 2 *Citations are not appropriate. The authors appear overly focused on citing high-impact factor articles, neglecting relevant ones. Neither the Heterocycles nor TL papers mentioned above are cited. Additionally, the authors stated, "However, the direct modification of valuable core structures by replacing one atom in a ring system without changing the ring size and*

aromaticity remains elusive, although it has been recognized as a highly desirable transformation”.

The direct modification of valuable core structures by replacing one or two atoms in a ring system without changing the ring size and aromaticity remains has been reported continuously in recent years. The present study is deeply interconnected with this literature and should be cited.

- *J. Am. Chem. Soc.* 129, 12640

- *Adv. Synth. Catal.* 355, 2495–

- *Chem. Commun.* 55, 8575

- *Org. Lett.* 23, 6126

- *Chem. Sci.* 14, 485

- *Nat. Chem.* doi.org/10.1038/s41557-023-01428-2

Response: We sincerely appreciate the valuable comments. These citations are of great help for us to describe the development of single-atom skeletal editing, and we have added them to the reference list.

Nat. Chem. doi.org/10.1038/s41557-023-01428-2 (see, ref. 26), Chem. Sci. 14, 485 (see, ref. 27), Org. Lett. 23, 6126 (see, ref. 28), Chem. Commun. 55, 8575 (see, ref. 29), Adv. Synth. Catal. 355, 2495 (see, ref. 30), J. Am. Chem. Soc. 129, 12640 (see, ref. 31).

3 *The quality of some data is unacceptable.*

The following NMR spectra exhibit significant levels of by-product contamination. Consequently, they may overestimate yields. It is recommended to purify the product sufficiently to obtain a clean spectrum before determining the yield, or alternatively, quantify the impurities via GC or other means to obtain accurate yields.

Compounds 15, 16, 19 (DCM?), 20, 38 (EtOAc?)

Response: We sincerely thank the reviewer for point out this issue. The clean spectrum of compounds 15, 16, 19, 20 and 38 has been added in SI on pages S93, S94, S97, S98 and S116 respectively. We also determined the yield using NMR analysis, employing 1,2-dichloroethane as an internal standard. The obtained yield closely corresponds to our isolated results.

Supplementary Figure 101. ¹H-NMR of compound **15**, recorded at 400 MHz and 25 °C in CDCl₃

NMR yield 65%

Isolated yield 62%

Supplementary Figure 103. ¹H-NMR of compound **16**, recorded at 400 MHz and 25 °C in CDCl₃

NMR yield 61%

Isolated yield 57%

Supplementary Figure 109. $^1\text{H-NMR}$ of compound **19**, recorded at 400 MHz and 25 °C in CDCl_3

NMR yield 59%

Isolated yield 52%

Supplementary Figure 111. $^1\text{H-NMR}$ of compound **20**, recorded at 400 MHz and 25 °C in CDCl_3

NMR yield 83%

Isolated yield 76%

Supplementary Figure 147. $^1\text{H-NMR}$ of compound **38** recorded at 400 MHz and 25 °C in $\text{DMSO-}d_6$

NMR yield 60%

Isolated yield 55%

Responses to the reviewer 2's comments

- 1 *In the proposed manuscript, authors describe an elegant skeletal editing strategy involving a straightforward carbon-to-nitrogen transmutations in arenols through a one-pot ring expansion-contraction sequence. This strategy avoided the dangers and laborious preparation of aryl azides as raw material, while using the readily available arenols as substrate with $N(nBu)4N3$ as reagent is a skillful strategy. I believe this report will inspire more researchers to develop other more perfect skeletal editing protocols and it will be bound to draw more researchers to apply the skeletal editing. In my own opinion, the scientific novelty of the manuscript is very high, and the quality of the work is high. My only critique is about the scale-up synthesis. I am satisfied with it too. My recommendation is to publish the manuscript.*

Response: We sincerely appreciate the valuable comments. A scale-up experiment (1.0 mmol **1a**) was carried out, gave **1** in 61% yield. This result has been added in Table 1.

Responses to the reviewer 3's comments

- 1 *The manuscript shows quite a bit of scope, except in the ortho substituent. The extent of scope in this regard is product 2 and product 12, which swap methyl for cyclopropyl and *n*-octyl. The authors should add a small set of additional substrates to explore the scope outside simple alkyl groups (Ph, CO₂Me, etc). Even if the results are negative, this would be informative to readers in terms of utility of this reaction.*

Response: We sincerely appreciate the valuable comments. As the author suggested, we have conducted an investigation on the substrate incorporating ester group (CO₂Me, CO₂Ph), oxime group, and phenyl. Under the standard conditions, only the substrate containing phenyl yielded the desired phenyl(quinolin-2-yl)methanone in 7% yield with a complex reaction system. However, decomposition was observed for other substrates. These findings have been included in the Supplementary Information on page S9, S43 and S125. I think the possible reason is that the conjugated system stabilizes the seven-membered ring intermediate, making it difficult to open the ring. This results in the occurrence of side reactions.

2 The authors should comment more clearly on the reaction of 44 to give a roughly 1:1 mixture of product 11 and deazidated 11a – what is the pathway of the deazidation?

Response: We sincerely thank the reviewer for pointing out this issue. Based on the literature (ARKIVOC 2009, 270-290), we hypothesized that the deazidation pathway proceeds through a sequential single-electron transfer (SET) process, ultimately affording in the formation of anion **I**. Subsequently, the loss of an azide ion from anion **I** would occur, leading to the production of **11a**. The single electron transfer of Fe³⁺ with naphthol may regenerate Fe²⁺ for further reaction. The corresponding literature has been added as reference 58.

- 3 *There are repeated typos referring to tricyclohexylphosphine oxide as tricyclohexylphosphine, both in the methods section of the manuscript and in the SI.*

Response: We sincerely appreciate the valuable comments. The tricyclohexylphosphine has been corrected to tricyclohexylphosphine oxide in the methods section of the manuscript and SI

- 4 *Additional referencing to N-atom insertion of nitro groups into arenes, accessing structurally similar 7-membered intermediates as in this paper, should be included. Work from Leonori (10.1002/anie.202310540, 10.1038/s41557-023-01429-1) and Radosevich (10.1021/jacs.2c12450) come to mind. The early work of Sundberg (10.1021/ja00757a032, 10.1021/jo00828a067, 10.1021/ja01031a024) should also be cited for appropriate context.*

Response: We sincerely appreciate the valuable comments. These citations are of great help for us to exhibit the application of 7-membered intermediates, and we have added them to the reference list.

10.1002/anie.202310540 (see, ref. 43), 10.1038/s41557-023-01429-1 (see, ref. 44), 10.1021/jacs.2c12450 (see, ref. 45), 10.1021/ja00757a032 (see, ref 46), 10.1021/jo00828a067 (see, ref. 47), 10.1021/ja01031a024 (see, ref. 48).

REVIEWERS' COMMENTS

Reviewer #3 (Remarks to the Author):

The authors appropriately addressed my concerns in their response letter. The manuscript is appropriate for publication in Nature Communications with two small edits. I simply request the authors put some of the detail they included in their letter into the manuscript. Please include the CO₂Me substrate yielding 0% product and Ph substrate yielding 7% product in Table 2 – this would be very informative for readers. Further, a brief mention of the presumed pathway for the deazidation should be included in the manuscript; I recommend “which demonstrated that the proposed azidation is a reversible process via successive SET from FeII to eliminate azide.[Ref 58]” Including the proposed mechanism of this deazidation in the SI in “Section 2.4 Control experiment of carbon-nitrogen transmutation” immediately following the reaction of 44 to yield 11 and 11a would make this more accessible to readers.

Responses to the reviewer's comments

- 1 *I simply request the authors put some of the detail they included in their letter into the manuscript. Please include the CO₂Me substrate yielding 0% product and Ph substrate yielding 7% product in Table 2 – this would be very informative for readers.*

Response: We really appreciate the valuable comments. The results of CO₂Me and Ph substrates have been added in Table 2 (now Figure 2).

- 2 *Further, a brief mention of the presumed pathway for the deazidation should be included in the manuscript; I recommend “which demonstrated that the proposed azidation is a reversible process via successive SET from Fe(II) to eliminate azide.[Ref 58]”. Including the proposed mechanism of this deazidation in the SI in “Section 2.4 Control experiment of carbon-nitrogen transmutation” immediately following the reaction of 44 to yield 11 and 11a would make this more accessible to readers*

Response: We really appreciate the valuable comments. According to the reviewer's suggestion, the brief mention “which demonstrated that the proposed azidation is a reversible process via successive single-electron transfer (SET) from Fe(II) to eliminate azide” has been included in manuscript. Furthermore, the proposed mechanism of deazidation has been included in the SI in “Section 2.4 Control experiment of carbon-nitrogen transmutation” following the reaction of 44 to yield 11 and 11a.